# Attitude Solving Algorithm and FPGA Implementation of Four-Rotor UAV Based on Improved Mahony Complementary Filter

**DOI:** 10.3390/s22176411

**Published:** 2022-08-25

**Authors:** Yanping Zhu, Jing Liu, Ran Yu, Zijian Mu, Lei Huang, Jinli Chen, Jianan Chen

**Affiliations:** 1School of Electronics and Information Engineering, Nanjing University of Information Science and Technology, Nanjing 210044, China; 2School of Networks and Telecommunications Engineering, Jinling Institute of Technology, Nanjing 211199, China

**Keywords:** UAV, attitude solving, Mahony complementary filtering, Allan variance, FPGA

## Abstract

With the development of modern industry, small UAVs have been widely used in agriculture, mapping, meteorology, and other fields. There is an increasing demand for the core attitude-solving algorithm of UAV flight control. In this paper, at first, a novel attitude solving algorithm is proposed by using quaternions to represent the attitude matrix and using Allan variance to analyze the gyroscope error and to quantify the trend of the error over time, so as to improve the traditional Mahony complementary filtering. Simulation results show that the six-axis data from the initial sensors (gyroscope and accelerometer) agree well with the measured nine-axis data with an extra magnetometer, which reduces the complexity of the system hardware. Second, based on the hardware platform, the six-axis data collected from MPU6050 are sent to FPGA for floating-point operation, transcendental function operation, and attitude solution module for processing through IIC communication, which effectively validates the attitude solution by using the proposed method. Finally, the proposed algorithm is applied to a practical scenario of a quadrotor UAV, and the test results show that the RMSE does not exceed 2° compared with the extended Kalman filter method. The proposed system simplifies the hardware but keeps the accuracy and speed of the solution, which may result in application in UAV flight control.

## 1. Introduction

UAV technology as the synthesis of many key technologies is widely used in many fields, such as remote sensing and mapping, aerospace, agriculture and environmental protection, road surface information extraction, and target tracking [1,2,3,4,5]. Attitude solving is one of the key techniques of the UAV flight control system [6]. Inertial measurement units (IMU) provide short-term position and orientation changes with low cost, low weight, and low power consumption. The gyroscope, accelerometer, and magnetometer are the main micro-electro-mechanical system (MEMS) sensors to realize stable flight of UAV. These sensors suffer from many sources of error, including high levels of noise, bias, axis misalignment, scale factor, and local temperature [7]. As a matter of fact, the flexibility and stability of the UAV flight are associated with the accuracy of the attitude solution and the quality of the hardware system [8].

The UAV sensor data fusion algorithm is widely used to improve the accuracy of UAV attitude solving. The commonly used data fusion algorithms are the extended Kalman filter (EKF), Mahony filter, and gradient descent methods [9,10,11]. Sabatini employed the EKF method to compensate for errors and obtain the attitude angle by solving quaternions [12]. However, the Kalman equation suffers from bias in the state estimation during linearization and requires a long recovery time. Mahony proposed a coupled nonlinear complementary velocity-assisted attitude filter that provides estimates of linear velocities in inertial and body coordinate systems [13]. However, the high noise in the body coordinate system affects the error analysis in attitude estimation. A dispersive quaternion-based controller for angular velocity measurement in spacecraft formation attitude was reported in Ref. [14], which can guarantee that each spacecraft achieves attitude synchronization when it is approaching the desired time-varying attitude and angular velocity during the formation maneuver. However, this method is only demonstrated in theoretical analysis without realistic UAV tests. Mayhew proposed a quaternion-based hybrid feedback scheme for three cases of attitude tracking problems, which can reduce the sensitivity of the system to noise while avoiding the “unwinding phenomenon” [15]. However, the disturbances and errors in the hardware system are not involved. An improved adaptive EKF algorithm was reported in Ref. [16] and the flight experiments to reduce the influence of noise and vibration disturbances on the attitude solution were carried out in UAV flight control. The experimental data and analysis show that the algorithm has significantly reduced the root-mean-square error (RMSE) of pitch and roll angles and improved the accuracy and stability of the flight attitude angle solution. In Ref. [17], a SINS-based adaptive multi-sample rotation vector attitude scheme to detect and correct non-exchangeability errors by analyzing the continuous gyroscope output and determining the threshold value is proposed. The simulation results show that the algorithm has good performance under the violent cone rotational motion. However, the results of the algorithm are not compared with that of the other inertial sensor calibration gyroscope. For the hardware platform of attitude solving, most investigations are based on STM and DSP processors. In Ref. [18], the attitude solving algorithm is based on quaternion, and Kalman Filter is used for data fusion to improve the accuracy by using a DSP processor. The system has an initial sensor and magnetic sensor, which generates nine-axis data. Compared with six-axis data, they are of higher complexity. Recently, the FPGA dual IP core platform is adopted as the data processor in the attitude solving of UAV [19]. The hardware design of the nine-axis attitude calculation is carried out based on an intelligent PID controller and complex EKF. The system also adds GPS and altimeter, which present high requirements for the hardware implementation algorithm and platform.

In this paper, to improve the accuracy and reduce the software and hardware requirements of the system, an FPGA-based UAV attitude solving system is designed and implemented. Considering the error sources of inertial sensors and FPGA platform characteristics, Mahony complementary filtering based on Allan variance with six-axis data fusion is proposed to compensate for the errors. Through practical verification, the results are very close to the results of nine-axis data solving, and the complexity of this algorithm is close to that of Kalman Filtering. In general, the proposed algorithm achieves high accuracy and strong real-time performance with a simplified hardware platform. The main contribution of this paper includes several aspects:(1)Mathematical concepts and formulas in attitude solving are introduced to provide the theoretical basis for the key steps in the attitude solving system, such as the representation of the attitude matrix, the derivation of the theoretical field vectors, and the quaternion update of the attitude kinematic equations.(2)Based on Allan variance, an improved Mahony complementary filtering using IMU six-axis data is proposed. Compared with the nine-axis data solving methods, i.e., Mahony and Madgwick, the accuracy of the proposed algorithm is comparable but has lower complexity.(3)The proposed algorithm is implemented on an FPGA platform. There are three steps to build the attitude solving system on the FPGA platform: data reading, data processing, and data sending, among which data processing as the focus of attitude solving contains key elements, such as floating-point operation, transcendental function operation, and system framework. The error analysis of the actual UAV test results is carried out.

The rest of this paper is organized as follows. The basic attitude solving principle and mathematical model are introduced in Section 2. In Section 3, an improved Mahony complementary algorithm based on Allan variance is proposed. The implementation of the hardware platform for attitude calculation is described in Section 4. The FPGA and UAV platform test results are also presented in this section. Finally, this paper concludes with Section 5.

## 2. Principle of Attitude Solution

### 2.1. Common Coordinate Systems and Representation of Attitude Matrix

The flight motion of a UAV can be decomposed into circular rotation around a fixed point and linear movement of coordinate positions. To investigate the attitude of UAV, it is necessary to choose a suitable coordinate system to describe the position, rotation speed, linear motion speed, and other physical quantities in the flight motion. There are two coordinate systems commonly used in engineering to describe attitude motion, the navigation coordinate system (*n*-system) and the body frame coordinate system (*b*-system), as shown in Figure 1. In fact, the angles between the two coordinate systems are the angular changes of the object after its motion [20]. In the body coordinate system, the corresponding angles of rotation about *X*, *Y*, and *Z* axes are roll angle φ, pitch angle θ, and yaw angle ψ, respectively.

The attitude solution is involved with the transformation between the *n* and *b* systems. The transformation between the two coordinate systems is essentially a vector transformation, which can be expressed by the cosine function between the vectors, so the attitude matrix can be called the directional cosine matrix. The Euler angles and quaternions are also optimized and transformed based on the direction cosine matrix.

The angle between the *n*-coordinate system and the *b*-coordinate system is called Euler’s angle. According to Euler’s theorem, a complex rotation can be decomposed by rotating a specific angle around *X*, *Y*, and *Z* axes, respectively, and the corresponding direction cosine matrix is expressed as *C*_1_, *C*_2_, and *C*_3_, respectively. Different rotation order demands different transformation formulas [21], and the general transformation is to rotate in the order of *Z-Y-X* axis. Then, the direction cosine matrix is expressed in terms of Euler angles as follows in Equation (1), where ψ, θ, φ are corresponding to yaw, pitch, and roll angle changes, respectively.
(1)$$C_{n}^{b}=\left[\begin{array}{ccc} \cos\theta \cos\psi & \cos\theta \sin\psi & -\sin\theta \\ \sin\varphi \sin\theta \cos\psi -\cos\varphi \sin\psi & \sin\varphi \sin\theta \sin\psi +\cos\varphi \cos\psi & \sin\psi \cos\theta \\ \cos\varphi \sin\theta \cos\psi +\sin\varphi \sin\psi & \cos\varphi \sin\theta \sin\psi -\sin\varphi \cos\psi & \cos\varphi \cos\theta \end{array}\right]\;\;\;\;$$

The theoretical gravitational acceleration $V_{b}$ in the body frame coordinate system can be derived from Equation (1):(2)$$V_{b}=C_{n}^{b}\cdot g_{n}=\left[\begin{array}{c} -\sin\theta g\\ \sin\psi \cos\theta g\\ \cos\varphi \cos\theta g \end{array}\right]$$
where $g_{n}=\left[\begin{array}{c} 0\\ 0\\ 1 \end{array}\right]$ is the unit vector of gravity direction of the *n*-system, and *g* is the gravity acceleration. The theoretical gravitational acceleration $V_{b}$ can compensate the gyroscope error well in the attitude solution, which is of great importance.

### 2.2. Equations of Attitude Kinematics Based on Quaternions

According to Euler’s theorem, an object moves in a circle around a fixed point, and the rotation process can be obtained by rotating the object around a fixed axis at a certain angle, and the fixed axis must pass the fixed point of the rotation of the circular motion [22]. The theoretical basis of this theorem is the property of the orthogonal matrix: there must exist a unit vector ***E*** such that ***E* = *CE***. This property indicates that the vector ***E*** representing the direction of the rigid body rotation axis has the same components in the rigid body coordinate system and the reference coordinate system [23]. Therefore, the attitude can be described by the following parameters, i.e., the three coordinate components of the unit vector of the rotation axis $e_{x}$, $e_{y}$, $e_{z}$, and the rotation angle θ.

The pose quaternion is defined as $q=q_{0}+q_{1}i+q_{2}j+q_{3}k.$ Individual parameters of the quaternion are not independent of each other and satisfy the following constraint relations [24].
(3)$$\left\{\begin{array}{c} q_{0}^{2}+q_{1}^{2}+q_{2}^{2}+q_{3}^{2}=1\\ \begin{array}{c} i^{2}=j^{2}=k^{2}=-1\\ \begin{array}{c} i=jk=-ij\\ j=ki=-kj\\ k=ij=-ji \end{array} \end{array} \end{array}\right.$$

The scalar part of the quaternion is $q_{0}=\cos\frac{\theta }{2}$, and the vector part is $q_{1}i+q_{2}j+q_{3}k=E\sin\frac{\theta }{2}$. According to the definition and physical meaning of quaternions: $q_{1}=e_{x}\sin\frac{\theta }{2}$, $q_{2}=e_{y}\sin\frac{\theta }{2}$, $q_{3}=e_{z}\sin\frac{\theta }{2}$. Therefore, the attitude matrix can be expressed in terms of quaternions as:(4)$$C_{n}^{b}=\left[\begin{array}{cc} q_{0}^{2}+q_{1}^{2}-q_{2}^{2}-q_{3}^{2} & \begin{array}{cc} 2\left(q_{1}q_{2}+q_{0}q_{3}\right) & 2\left(q_{1}q_{3}-q_{0}q_{2}\right) \end{array}\\ \begin{array}{c} 2\left(q_{1}q_{2}-q_{0}q_{3}\right)\\ 2\left(q_{1}q_{3}+q_{0}q_{2}\right) \end{array} & \begin{array}{cc} \begin{array}{c} q_{0}^{2}-q_{1}^{2}+q_{2}^{2}-q_{3}^{2}\\ 2\left(2q_{3}-q_{0}q_{1}\right) \end{array} & \begin{array}{c} 2\left(2q_{3}+q_{0}q_{1}\right)\\ q_{0}^{2}-q_{1}^{2}-q_{2}^{2}+q_{3}^{2} \end{array} \end{array} \end{array}\right]=\left[\begin{array}{cc} T_{11} & \begin{array}{cc} T_{12} & T_{13} \end{array}\\ \begin{array}{c} T_{21}\\ T_{31} \end{array} & \begin{array}{cc} \begin{array}{c} T_{22}\\ T_{32} \end{array} & \begin{array}{c} T_{23}\\ T_{33} \end{array} \end{array} \end{array}\right]$$

According to the definition of quaternion, the attitude at a certain moment in the motion can be expressed as:(5)$$Q=\cos\frac{\theta }{2}+E\sin\frac{\theta }{2}$$

Equation (5) is the attitude kinematic equation, differentiated for time, which can be obtained from $E\cdot E=-1,\;\frac{dE}{dt}=0$:(6)$$\frac{dQ}{dt}=\frac{1}{2}\cdot E\cdot \frac{d\theta }{dt}\cdot Q$$
where $E\cdot \frac{d\theta }{dt}$ is equal to angular velocity ω. Considering the implementation on an FPGA hardware platform, the solution of this differential equation by the first-order Runge–Kutta method here yields:(7)$${\left[\begin{array}{c} q_{0}\\ q_{1}\\ q_{2}\\ q_{3} \end{array}\right]}_{t+dt}={\left[\begin{array}{c} q_{0}\\ q_{1}\\ q_{2}\\ q_{3} \end{array}\right]}_{t}+\frac{dt}{2}\left[\begin{array}{c} -\omega_{x}q_{1}-\omega_{y}q_{2}-\omega_{z}q_{3}\\ \omega_{x}q_{0}+\omega_{z}q_{2}-\omega_{y}q_{3}\\ \begin{array}{c} \omega_{y}q_{0}-\omega_{z}q_{1}+\omega_{z}q_{3}\\ \omega_{z}q_{0}+\omega_{y}q_{1}-\omega_{x}q_{2} \end{array} \end{array}\right]$$

Equation (7) is the quaternion update equation in the attitude solution [25], where $\omega ={\left[\omega_{x}\;\omega_{y}\;\omega_{z}\right]}^{T}$. The angular velocity can be converted into quaternions. Using quaternions for attitude solution, there is no complicated trigonometric function solution, which greatly reduces the computational effort, and there is no singularity phenomenon, which can realize the full attitude solution, and the method plays an important role in the subsequent design of the attitude solution algorithm. It is an important theoretical basis for updating the attitude kinematic equations of the algorithm designed in this paper.

## 3. Design and Simulation of the Attitude Solution Algorithm

### 3.1. Inertial Sensor Error Analysis

The inertial sensor used in this design is MPU6050, and its output is six-axis data. The data acquisition unit consists of a gyroscope for measuring angular velocity and an accelerometer for measuring acceleration, and the data processing of attitude solving is done by the FPGA platform.

In the hardware principle, it is known that the gyroscope measures the angular velocity, and the angle can be obtained directly for the integration; the accelerometer measures the projection of the gravitational acceleration on the three axes of the ***b*** system, and the angle can be found by trigonometric functions. According to the working principle of the gyroscope and accelerometer, the angle calculation formula can be derived, taking the roll angle as an example.
(8)$$\left\{\begin{array}{c} roll_{gyro}=\int \omega dt\\ roll_{acc}=arctan\frac{a_{x}}{-a_{z}} \end{array}\right.$$

According to Equation (8), the angle is solved in MATLAB with the actualmeasured three-axis angular velocity and three-axis acceleration as shown in Figure 2, and the sampling frequency for the measurement is f=200 Hz, unit time is $dt=\frac{1}{f}=0.005\;s$.

From the above curves, the error sources of MPU6050 can be summarized as the following two points.

①The gyroscope’s zero bias is serious, that is, the gyroscope has angular velocity output even when it is stationary, which is amplified by integrating zero bias, resulting in serious angular drift over time. Zero bias can be seen as a low-frequency noise with large amplitude and slow change.②The accelerometer has good low-frequency characteristics, but there is high-frequency noise, and the operating principle causes the accelerometer to be unable to sense horizontal rotation, so the measured yaw angle accuracy is insufficient.

### 3.2. Improved Mahony Complementary Filtering Based on Allan Variance

According to the error characteristics of gyroscope and accelerometer, there can be different filtering algorithms for error reduction, such as Kalman filter, linear complementary filter and adaptive complementary filter, etc. The computational complexity of the complementary filter is significantly smaller than that of the Kalman filter, and less dependent on the implementing hardware platform. As a result, the complementary filter can effectively reduce the design difficulty of the system and promote a realistic application. Now, many small UAVs use the nonlinear Mahony complementary filtering algorithm, which is different from the above filtering algorithm when Mahony complementary filtering does not start from the perspective of the error frequency, but from the perspective of the physical meaning of the error for error compensation. After the inertial sensor data are measured, the theoretical field vector is derived by updating the quaternion according to Equation (7), and the actual field vector measured by the six-axis sensor and the theoretical field vector are quaternion multiplication. According to the definition of vector quaternion multiplication, the final result and the error angle satisfy the triangular function relationship, so the gyroscope can be corrected with this error.

The core of Mahony complementary filtering is to use the quaternion multiplication of the theoretical vector and the actual vector to calculate the error and then correct the gyroscope. The error is then compensated to the gyroscope using the PI controller in the PID algorithm. However, Mahony still has some disadvantages. For instance, in some practical application scenarios, the complex magnetic field environment will make the magnetometer error very large, resulting in nine-axes data after complementary filtering, which is not as good as six-axes data, but if only giving up the magnetometer, the accelerometer cannot correct the yaw angle.

To overcome the shortcomings of the Mahony algorithm, a complementary filtering algorithm is proposed in this part to analyze the zero bias of the gyroscope using Allan variance. The role of both accelerometer and magnetometer in the principle of the Mahony algorithm is to compensate for the zero bias of the gyroscope according to the attitude of the current moment. The reason for real-time compensation is that the zero bias of the gyroscope is not a constant value, it will change in real-time with the moving process, and if it is possible to quantify this change. It can compensate the yaw angle to some extent instead of the error measured by the magnetometer.

Allan variance is a common error analysis method for analyzing each noise of the gyroscope, because of its low algorithmic complexity and small computational effort, and it can quantify each error coefficient by dividing the power spectrum uniformly [26]. It should be noted that the use of Allan variance requires a long period of gyroscope stationary angular velocity data, the reason why a long period of stationary time is required is that the gyroscope zero bias is not a fixed value, its value will change over time, and the variable quantity is different in all three axes. Allan variance can converge this magnitude of change to a fixed value by analyzing a long period of data, which is called zero bias instability.

Among the noise of Allan’s variance, the zero-bias instability [27] is an important indicator of gyroscope performance. The zero bias instability ***B*** represents the magnitude of the change in zero bias ***d*** over a period of time in units °/h, so the error can be written as:(9)$$\delta^{2}=d+\frac{B}{3600}\times dt\times \mathrm{Time}$$

Ideally, the gyroscope with Allan variance is shaped like the hook in Figure 3a, in which the zero bias instability ***B*** is the lowest point of the curve.

The official gyroscope data of PX4 is analyzed with Allan variance. Since the gyroscope has fewer data at rest, the analyzed data is somewhat different from the theoretical situation as shown in the Figure 3b. The zero bias instability is 0.558054 on X-axis, 2.620662 on Y-axis, and 5.506688 in Z-axis.

Using Allan’s variance to quantify the zero bias and zero bias instability of the gyroscope, the zero bias error can be calculated according to Equation (7), which is used to replace the error of the magnetometer. The process of the Mahony complementary filter method based on Allan variance is shown in Figure 4. Step 1:Gyroscope and accelerometer measure angular velocity
ω and acceleration a respectively;Step 2:According to ***α*** = ***V**_b_* ⊗ ***V**_n_*, calculate the accelerometer error ***α***; Based on Formula (9), calculate Allan variance ***δ***^2^. The system error vector is $e=\alpha +\delta^{2}$;Step 3:Use the error vector as an input to the PID controller. Obtain the correction value for the angular velocity: $\Delta \omega =(K_{p}+K_{f}\cdot 1/s)\;e$;Step 4:Calculate the corrected angular velocity vector: $\omega^{\prime }=\omega +\Delta \omega$;Step 5:Update the quadratic ***q*** using Equation (7), and then update ***V***_b_ using Equation (2);Step 6: Start a new round of calculations from step 1.

Based on Mahony’s complementary filter, Allan variance is used to correct the Gyroscope error to improve the accuracy of the six-axis attitude estimation. The following part is the simulation comparison.

### 3.3. Simulation of Attitude Solution

Improved Mahony algorithm simulation is based on PX4 official six-axis data, and traditional Mahony algorithm and Madgwick algorithm are based on PX4 nine-axis data. Six-axis data parameters are angular velocity and acceleration, and nine-axis data are based on six-axis with the magnetometer’s three-axis data. The simulation parameters are set to be: *K_p_* and *K_i_* of PID controller are 0.001 and 0.1, respectively, the unit time is 0.0039 s, and the zero-bias instability of the three axes obtained by Allan variance are 0.058, 0.049, and 0.061, respectively. The simulation results are shown in Figure 5, Figure 6 and Figure 7, and it can be found that the results simulated by the proposed algorithm agree well with the nine-axis data.

It can be seen that the noise and error of Madgwick’s algorithm are larger. Comparing the proposed algorithm with Mahony’s algorithm, the average absolute errors of roll, pitch, and yaw are 2.8778°, 0.9186°, and 3.3649°, respectively. The error of yaw angle is slightly larger than the error of roll and pitch angle, indicating that the role of the geomagnetic meter in correcting yaw angle is still better than Allan variance. However, it should be noted that, due to the short gyroscope stationary state in the official PX4 data, the zero bias instability using Allan variance is not representative, and the error of the three angles can be further reduced by actual measurement data. The analysis of this improved algorithm error by the sampling of the actual sensor from the hardware platform will be provided in Section 4.

## 4. Implementation of Attitude Calculation on FPGA Hardware Platform

### 4.1. FPGA System Implementation and Test

The designed attitude-solving hardware system consists of an AX1702 development board, MPU6050 inertial sensor, and the upper computer. The FPGA development board communicates with MPU6050 through the IIC serial port, and the attitude solving is performed on the FPGA development board. Finally, the attitude angle data are sent to the computer for display. Figure 8 shows the hardware system framework, and Figure 9 shows the solving system flow in detail. The core issues of the system implementation are: (1) floating point operation; (2) transcendental function operation; (3) attitude solving module.

Vivado has the function to configure floating-point IP cores according to the requirements, which makes the floating point module more efficient and standardized. This design uses the IEEE754 standard for symbol, exponent, and tail for logical operations. As the attitude solution involves the calculation of some transcendental functions, including square root, inverse tangent, and inverse chord, which involve the calculation of a large number of floating-point numbers, when the amount of calculation is small, the lookup table or level expansion of the operation can be used, but these two methods will consume large storage and hardware multiplication unit resources. It is difficult to save hardware at the same time to take into account the accuracy of the algorithm, so it is only suitable for the scenarios with a small computational amount. Compared with these two methods, the CORDIC algorithm has the advantages of increasing the operation speed and reducing the resource consumption: (1) CORDIC only needs a shifter and accumulator during the calculation, and does not need any hardware multiplication unit; (2) CORDIC algorithm also requires very little storage resources, and only needs a small number of storage resources for pre-storing data. In summary, this design uses the CORDIC algorithm to calculate the transcendental function.

The core attitude solving based on the FPGA platform is implemented by a combination of serial and parallel operations. The serial operation requires some pre-processing results: (1) the serial operation process of angular velocity and acceleration of inertial sensor data from the original processing to normalization and then update the error compensation; (2) define the value of the initial quaternions, use quaternions to calculate the rotation matrix and then extract the gravity component in the attitude matrix; (3) complementary filter, add the error into the PID controller with the angular velocity measured by the gyroscope, and correct the angular velocity value. After updating the error compensation, another serial operation is performed, from updating the angular velocity to updating the quaternion, then normalizing the quaternion, and finally converting the quaternion into a Eulerian angle output through the parallel CORDIC algorithm module. This combination of serial and parallel design satisfies the logic requirements of the algorithm and increases the processing speed of the system, which is the advantage of using FPGAs for attitude solving.

The experimental results are shown in Figure 10. Compared with the actual values, Roll. Pitch and Yaw angles have very high accuracy. The average error of Roll or Pitch angle is less than 0.2°, and the average error of Yaw is less than 2°.

### 4.2. UAV Measurement System

The actual test was carried out on Four-Rotor UAV PX4 450, and the flight trajectory is shown in Figure 11. The quadrotor UAV used in this experiment is composed of a mechanical system, a drive system and a control system through pixhawk open source, ESCs (Electronic Speed Controller) and motors. The UAV flies at different horizontal levels, and the track twists and turns. The attitude angle data after the attitude solution was imported into MATLAB for analysis, and the three attitude angles solved using the algorithm proposed in this paper were compared with the results of Mahony, Madgwick, and EKF algorithms [28,29,30] as shown in Figure 12, Figure 13 and Figure 14, respectively. All algorithms use the six-axis data as the input parameters. EKF has the best accuracy due to its higher complexity, and the accuracy of the proposed algorithm is better than the traditional Mahony and Madgwick algorithms. By analyzing the test data, the attitude resolution errors of roll, yaw and pitch data are within 2° at the beginning of flight. After steadily, the pitch and roll errors can be maintained within 0.2° in a relative long time. The proposed algorithm can achieve excellent accuracy and stability for an increased flight time. The error comparison data are shown in Table 1 and Table 2. The average absolute errors of the calculated angles are compared with the other three algorithms, as shown in Table 1. The maximum errors of the angles were calculated using MATLAB and compared with that reported in Ref. [31] and the product MTi-G-710 from the Dutch company Xsens Technology, as shown in Table 2. The proposed algorithm has the best accuracy in roll and pitch angles and similar accuracy of raw angle with the results reported in [31].

Based on the real-time requirements of the UAV flight control system, the algorithms are compared in terms of the time consumed, as shown in the following Table 3. This proposed method is the best in real-time implementation.

## 5. Conclusions and Discussion

The widespread use of UAVs demands the hardware of low-cost to process the attitude solving in the UAV flight control. Considering the high accuracy of the attitude solving algorithm and real-time implementation on hardware, a novel UAV system including an attitude solution algorithm and simplified hardware system is proposed. Based on the analysis of the error sources of inertial sensors, the Allan variance–Mahony complementary filtering algorithm is reported and its advantages are verified theoretically. The corresponding attitude-solving algorithm is implemented on an FPGA platform in real-time. The system is characterized by a reasonable and effective attitude solution accuracy and simple hardware construction, which has a smaller data processing volume when compared with the system using the nine-axis data. The disadvantage of the reported system is that although the Allan variance can analyze the Gyroscope zero bias instability, it needs a long time in stationary state to obtain more accurate analysis results. In realistic applications, the percentage of error in Allan variance analysis can be determined by the length of the stationary state. In future work, a more optimal mathematical model will be investigated to improve the accuracy of the attitude solution.

## Figures and Tables

**Figure 1 sensors-22-06411-f001:**
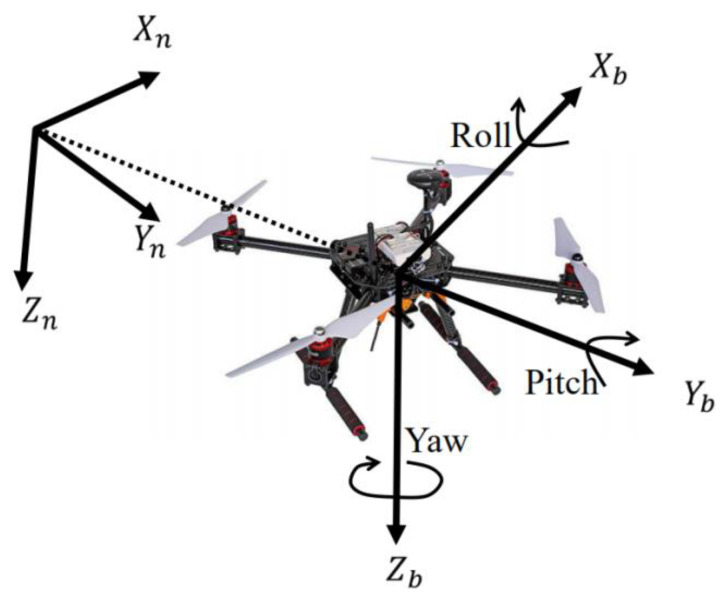
Two coordinate systems.

**Figure 2 sensors-22-06411-f002:**
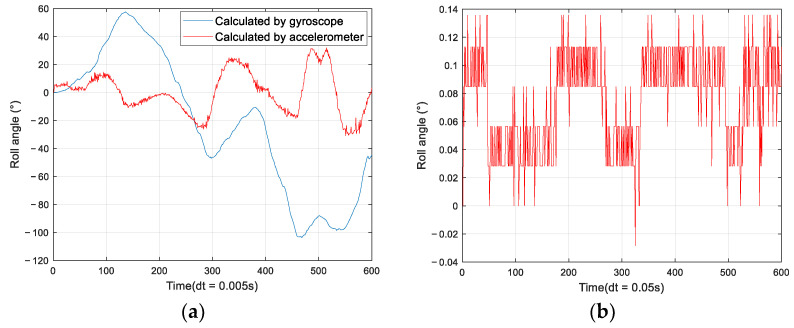
Error analysis. (**a**) Angular velocity and acceleration curves; (**b**) Gyroscope zero bias.

**Figure 3 sensors-22-06411-f003:**
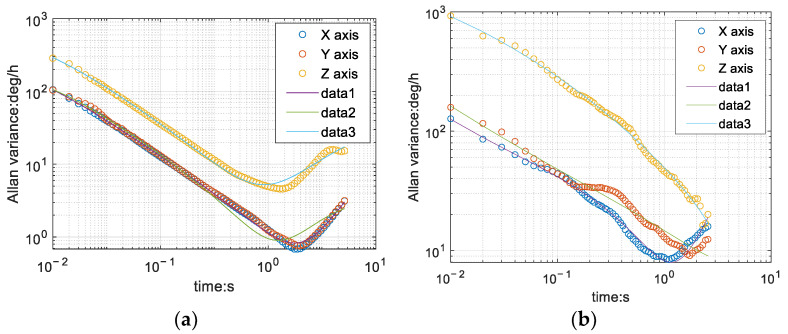
Allan variance analysis. (**a**) Theoretical value; (**b**) PX4 nine-axis data.

**Figure 4 sensors-22-06411-f004:**
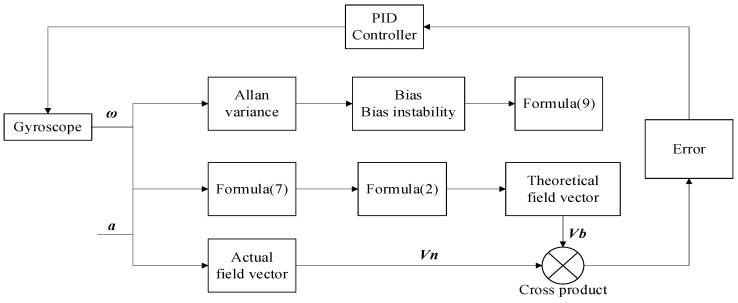
Block diagram of the improved Mahony algorithm using Allan variance.

**Figure 5 sensors-22-06411-f005:**
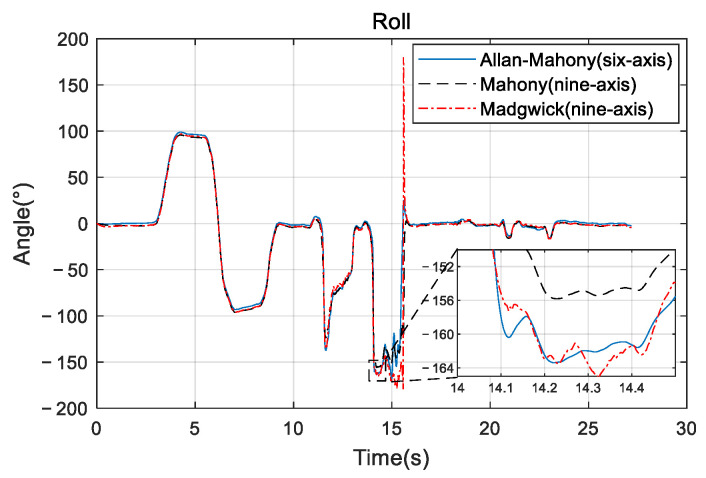
Roll angle comparison.

**Figure 6 sensors-22-06411-f006:**
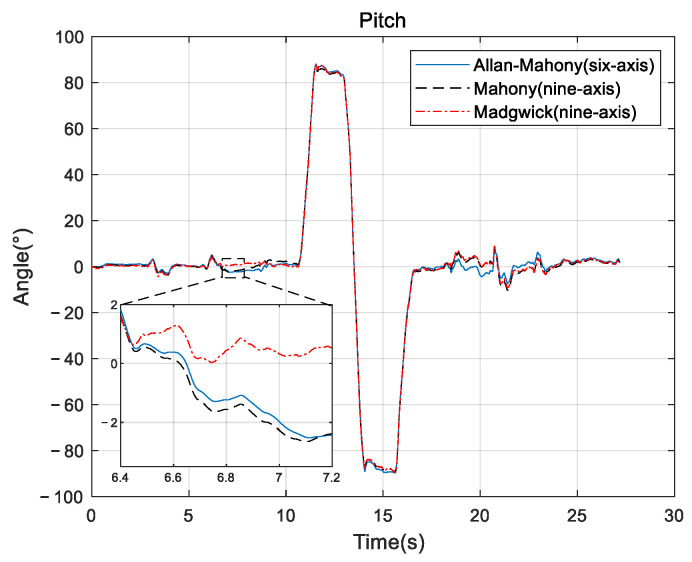
Pitch angle comparison.

**Figure 7 sensors-22-06411-f007:**
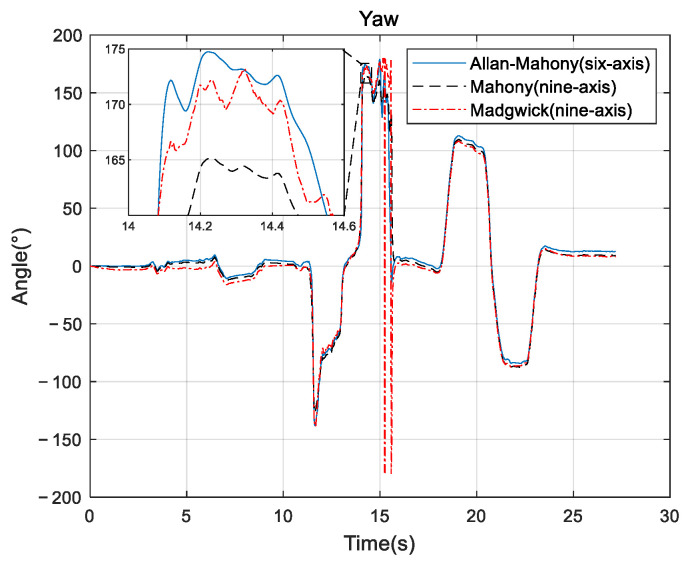
Yaw angle comparison.

**Figure 8 sensors-22-06411-f008:**
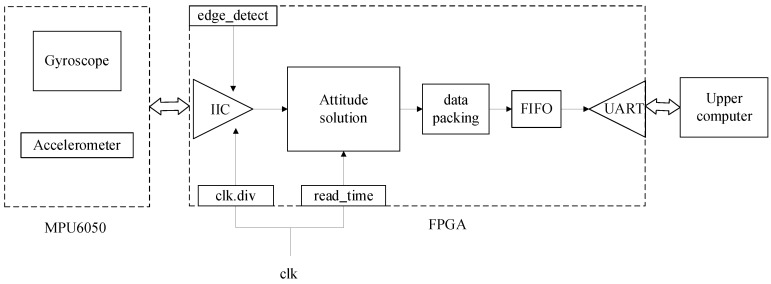
The framework of hardware system with FPGA.

**Figure 9 sensors-22-06411-f009:**
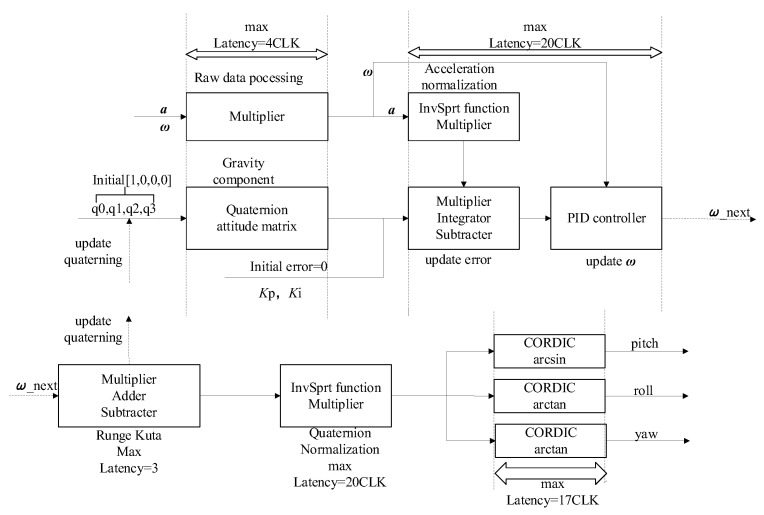
Attitude solution process on FPGA platform in detail.

**Figure 10 sensors-22-06411-f010:**
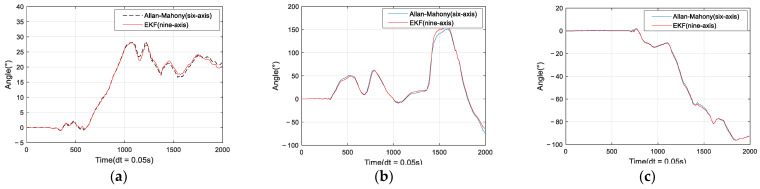
FPGA platform test results based on the proposed algorithm. (**a**) Roll; (**b**) Pitch; (**c**) Yaw.

**Figure 11 sensors-22-06411-f011:**
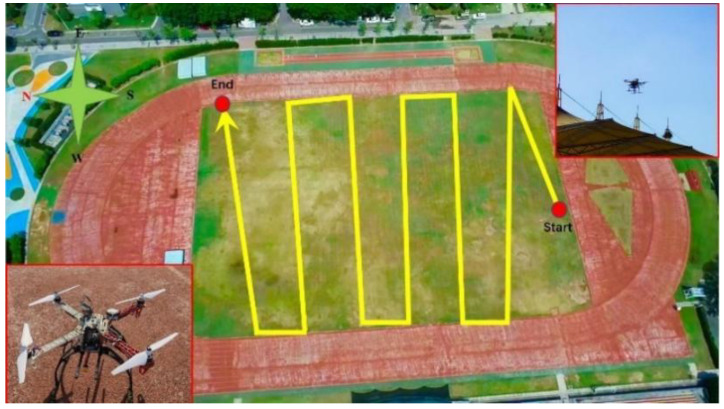
Flight Scenario and track.

**Figure 12 sensors-22-06411-f012:**
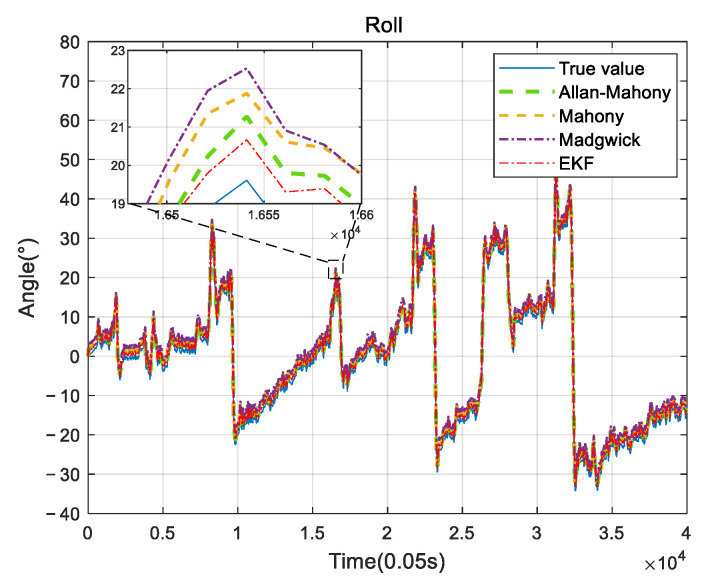
Roll comparison.

**Figure 13 sensors-22-06411-f013:**
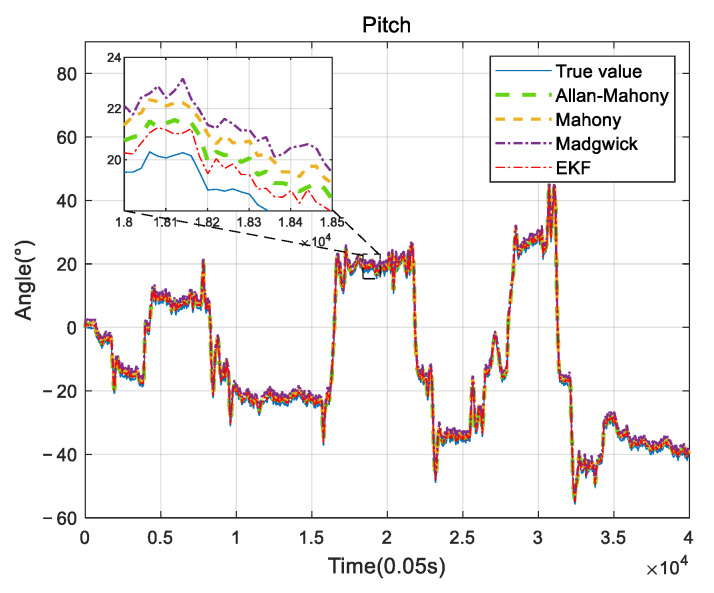
Pitch comparison.

**Figure 14 sensors-22-06411-f014:**
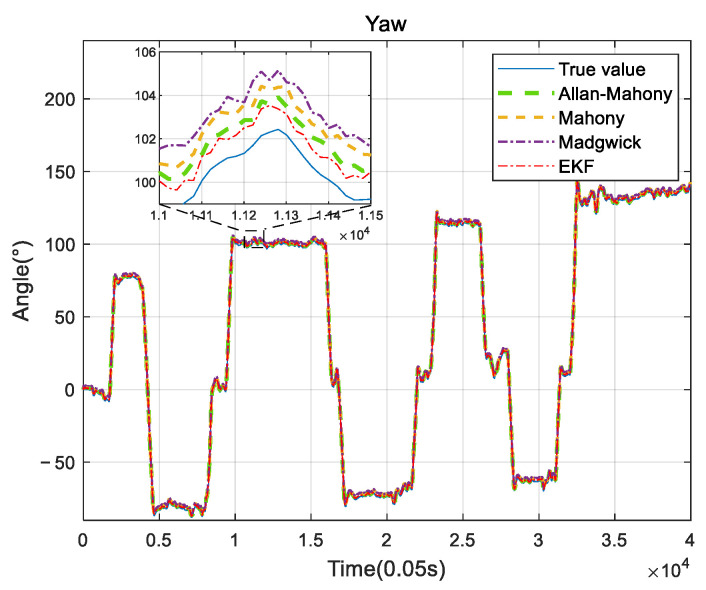
Yaw comparison.

**Table 1 sensors-22-06411-t001:** Comparison of mean absolute errors.

Average Error Data Source	Roll (°)	Pitch (°)	Yaw (°)
This paper	1.2383	0.8641	2.6764
Madgwick	2.4425	5.8539	5.8096
Mahony	2.1871	1.3685	5.1719
EKF	0.3195	0.3019	0.7315

**Table 2 sensors-22-06411-t002:** Maximum error comparison.

Maximum Error Data Source	Roll (°)	Pitch (°)	Yaw (°)
This paper	3.152	3.364	5.012
Ref. [31]	6	5	4.5
MTi-G-710	10	10	15

**Table 3 sensors-22-06411-t003:** Real-time comparison of different algorithms.

Algorithm	Time (s)
This paper	0.1568
Madgwick	0.2080
Mahony	0.1782
EKF	0.2895

## Data Availability

The datasets used to support the findings of this study are available from the corresponding author upon reasonable request.

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
