# Peer review of "Attitude Solving Algorithm and FPGA Implementation of Four-Rotor UAV Based on Improved Mahony Complementary Filter"

_sensors, 2022, doi:10.3390/s22176411_

Round 1
Reviewer 1 Report
In the sentence on line 262: ” Use the error vector as an input to the PID controller. obtain the correction value for the angular velocity:” is necessary to use capital letter in the first letter of sentence.
Better sharpness of picture 3.
The value Vn in the picture 4 is necessary to put out of the line.
In Fig. 5 and 6 less, circulate text with axes.
Increase the sharpness of the outlines and descriptions of the axes in the pictures.
Figure 4, 8, 9 indent from text.
Author Response
Accordind to the reviewer’s comments, I revised most of the problems, including references, grammar errors, further processing of some figures,brief introduction of UAV platform,and some analysis of results. Please see the attachment.

Reviewer 2 Report
The paper proposes a novel strategy to improve the accuracy and reduce the software and hardware requirements for UAV flight controller systems. The strategy consists of implementing an attitude solving algorithm and FPGA system for UAVs based on an Improved Mahony Complementary Filter.
The paper is generally well-written, and the proposed strategy is explained adequately, except for a few sentences that are hard to follow (see the annotated pdf). The mathematical part is appealing and looks well-clarified. The strategy seems to be suited for real-case scenarios and to work in real-time. The provided results seem interesting compared with other state-of-the-art procedures.
General remarks and suggestions:
• Line 29: UAVs research is not limited to only these fields nowadays. I suggest citing more recent works in other fields like:
1. Li, C. L., Sohn, K., Yoon, J., & Pfister, T. (2021). Cutpaste: Self-supervised learning for anomaly detection and localization. In Proceedings of the IEEE/CVF Conference on Computer Vision and Pattern Recognition (pp. 9664-9674).
2. Aslan, M. F., Durdu, A., Sabanci, K., Ropelewska, E., & Gültekin, S. S. (2022). A comprehensive survey of the recent studies with uav for precision agriculture in open fields and greenhouses. Applied Sciences, 12(3), 1047.
3. Biçici, S., & Zeybek, M. (2021). An approach for the automated extraction of road surface distress from a UAV-derived point cloud. Automation in Construction, 122, 103475.
4. Piramuthu, O. B., & Caesar, M. (2022, April). UAV-VANET authentication for real-time highway surveillance. In Proceedings of the 37th ACM/SIGAPP Symposium on Applied Computing (pp. 1925-1931).
5. Avola, D., Cinque, L., Diko, A., Fagioli, A., Foresti, G. L., Mecca, A., ... & Piciarelli, C. (2021). MS-Faster R-CNN: Multi-stream backbone for improved Faster R-CNN object detection and aerial tracking from UAV images. Remote Sensing, 13(9), 1670.
• All the Figures look to be misplaced with respect to the text. Moreover, some of them are not introduced in the textual part of the paper. The reference for the Figure in line 181 sounds to be Figure 2, not Figure 3. I suggest enhancing some Figures’ quality, for example, Figure 3 (also increasing the size for better readability), Figure 5, Figure 7, Figure 10 (also leveling out the legend and the axis aesthetic), Figures 12,13,14 (the colors are really similar, which impact the readability in color prints and make it completely unreadable in b/w). In general, I also recommend checking all the references for Figures and fixing misplacements.
• In Section 4.2 (lines 346-377) only one case scenario is reported. Does it correspond to the only test you had for your system? Even if the proposed scenario has different turns, it could be interesting to test the proposed method also with other trajectories to ensure its general robustness.
• Could you please provide more details about the hardware capabilities and equipment of the UAV used in the experiments?
• Is the proposed system UAV independent? To better explain my point: Do the errors you reported for both your strategy and the ones in the comparison change according to the flight length? This is crucial, because, nowadays, UAV equipment is boosting in quality, also allowing for an increased flight time.
• In my opinion, a point to highlight is: Is the proposed system entirely autonomous from take-off to landing as I intended from the text? In this case, writing a sentence in this regard could be useful.
Punctual observation and suggestion:
• Refer to the annotated pdf for punctual observations, typos, and hard-to-read sentences.
• In general, I also suggest grammar checking the paper to improve English quality and readability
Author Response
Accordind to the reviewer’s comments, I revised most of the problems, including grammar errors, further processing of some figures,brief introduction of UAV platform,and some analysis of results. Please see the attachment.
